# Multicountry research on comprehensive abortion policy implementation in Latin America: a mixed-methods study protocol

Celina Gialdini [1,2] Agustina Ramón Michel,[3] Mariana Romero,[3,4] Silvina Ramos,[3] Guillermo Carroli,[1] Berenise Carroli,[1] Rodolfo Gomez Ponce de León [5], Mercedes Vila Ortiz,[1,6] Antonella Lavelanet[7]

For numbered affiliations see end of article.

**Correspondence to**
Dr Celina Gialdini;
cgialdini@crep.org.ar

## ABSTRACT

**Introduction** Access to comprehensive abortion care could prevent the death of between 13 865 and 38 940 women and the associated morbidity of 5 million women worldwide. There have been some important improvements in Latin America in terms of laws and policies on abortion. However, the predominant environment is still restrictive, and many women, adolescents and girls still face multiple barriers to exercise their reproductive rights. This research will systematically assess comprehensive abortion policies in five Latin American countries (Argentina, Colombia, Honduras, Mexico and Uruguay). The aim is to identify barriers, facilitators and strategies to the implementation of abortion policies, looking at four key dimensions— regulatory framework, abortion policy dynamics, abortion service delivery and health system and health outcomes indicators—to draw cross-cutting lessons learnt to improve current implementation and inform future safe abortion policy development.

**Methods and analysis** A mixed-method design will be used in the five countries to address the four dimensions through the Availability, Accessibility, Acceptability and Quality of Care model. The data collection tools include desk reviews and semi-structured interviews with key actors. Analysis will be performed using thematic analysis and stakeholder analysis. A regional synthesis exercise will be conducted to draw lessons on barriers, facilitators and the strategies.

**Ethics and dissemination** The project has been approved by the WHO Research Ethics Review Committee (ID: A66023) and by the local research ethics committees. Informed consent will be obtained from participants. Data will be treated with careful attention to protecting privacy and confidentiality. Findings from the study will be disseminated through a multipurpose strategy to target diverse audiences to foster the use of the study findings to inform the public debate agenda and policy implementation at national level. The strategy will include academic, advocacy and policy arenas and actors, including peer-reviewed publication and national and regional dissemination workshops.

## STRENGTHS AND LIMITATIONS OF THIS STUDY

⇒ The study encompasses five countries with a variety of legal environments regarding abortion in order to show different contexts while looking at common patterns regarding barriers, facilitators and strategies.

⇒ From all complex social, legal, cultural and ethical factors shaping abortion care and policies, this research will focus on four, using the Availability, Accessibility, Acceptability and Quality of Care framework.

⇒ This study will focus on factors influencing implementation.

⇒ Data will be collected through standardised instruments, which might not reflect the heterogeneity among and within participating countries.

## INTRODUCTION

Access to comprehensive evidence-based abortion care could prevent the death of between 13 865 and 38 940 women[1] and the associated morbidity of 5 million women worldwide.[2] Improving access to quality abortion care is an essential strategy for the provision of universal access to reproductive health and for the achievement of the United Nations Sustainable Development Goals (SDG): good health and well-being and gender inequality (SDGs 3 and 5). Women's access to sexual and reproductive rights can also contribute to reproductive empowerment, addressing additional SDGs such as quality education (SDG 4), sanitation and hygiene (SDG 6), decent work and economic growth (SDG 8), reducing inequalities (SDG 10) and fostering peace, justice and strong institutions (SDG 16).[3]

Since the adoption of the Programme of Action of the International Conference on Population and Development 28 years ago, countries have been called on to strengthen their commitment to women's health by addressing unsafe abortion and supporting a woman's right to decide.[4] Although some improvements in terms of policies on sexual and reproductive health— including law reforms—have been achieved,

many women in Latin America face multiple barriers to exercising their reproductive rights.[5]

Obstacles to quality abortion access are a matter of human rights and of public health: in this region, at least 10% of all maternal deaths are a consequence of unsafe abortions and near 760 000 women are hospitalised per year due to complications related to unsafe abortion, which violates their rights to life and health.[6] The obstacles faced by women, adolescents and girls when trying to access quality abortion include the interpretation and the implementation of legal grounds (ie, legal indications under which is legal to have an abortion), the lack of access to information, stigma, restricted availability of healthcare providers and scarcity of facilities that can lawfully provide services, among others.[7] These barriers can disproportionately affect specific groups, such as adolescents, women that are poor, Indigenous women, women living in small towns and other women in vulnerable social conditions.

In this context, reforming abortion laws has proven to be challenging. Most Latin American countries have restrictive abortion laws, allowing legal termination of pregnancy only under certain grounds such as rape, severe fetal impairment or when the woman's life or health is at risk.[8] Abortion is legal without restriction as to reason in the first trimester of pregnancy in Argentina, Cuba, Uruguay and nine states of Mexico (Mexico City, Oaxaca, Hidalgo, Veracruz, Baja California, Baja California Sur, Sinaloa, Guerrero and Colima). Moreover, in February 2022, the Colombian Constitutional Court ruled to decriminalise abortion up to 24 weeks of pregnancy.

Despite a restrictive environment, some countries have managed to enhance the availability, accessibility, acceptability and quality of abortion services and care by resorting to alternative strategies. Court judgements and public health guidelines have been used to expand the interpretation of the circumstances under which abortion is allowed. In settings with very restrictive legal environments, like Honduras, guidelines to manage postabortion care have also been issued.[9]

Taking this regional context into consideration, we will systematically look at comprehensive abortion policies in five Latin American countries—Argentina, Colombia, Honduras, Mexico and Uruguay—to understand how context, content, actors and processes affect the implementation of comprehensive abortion policies. The aim is to assess abortion regulations (by *regulations* we mean rules or guidelines established and enforced by an authority, such as constitutional clauses, laws, decrees and other sort of administrative regulations) and their use in practice, the policy dynamics, the service delivery arrangements and the abortion-related health indicators and how they are monitored and evaluated, to identify barriers and facilitators to the implementation of comprehensive abortion policy and the strategies used to address them in each country. Finally, we will identify similarities and differences among barriers, facilitators and strategies of policy implementation across countries, and draw lessons

learnt from these implementation processes and the strategies to improve access to evidence-based abortion care.

## METHODS AND ANALYSIS
### Design and framework
In this research study we will look at the context, content, actors and processes affecting the implementation and monitoring of comprehensive abortion policies and comprises five country-case studies and a regional synthesis of findings. We will use a mixed-method design applying diverse data gathering techniques and analysis strategies to assess four key dimensions of abortion policy (by *policy* we mean a system of laws, regulatory measures, courses of action and funding priorities concerning a given topic promulgated, in this case, abortion.):

▶ The **regulatory framework** (dimension 1).
▶ The **policy dynamics** (dimension 2).
▶ The **service delivery arrangements** (dimension 3).
▶ The **health system and health outcomes indicators** (dimension 4).

In dimension 1 (regulatory framework) we will assess how and to what extent regulations—both written and in practice—pose as barriers and facilitators to quality abortion access, and the strategies to deal with them. We will address the traits and use of existing abortion-related regulations. This dimension encompasses laws, resolutions, decrees, health standards and health guidelines, technical guidelines and other official documents with or without legal status. First, we will perform a descriptive analysis of the regulatory framework, and then an examination of the practice.

In dimension 2 (policy dynamics) we will assess how aspects such as agenda setting, design and implementation processes and governance and actors' involvement, pose as barriers and facilitators of abortion policy and how they are addressed through different strategies. We will assess the agenda setting looking at the political and social conditions that place abortion policy on the public agenda. For design and implementation aspects we will focus on the shaping of content in policy formulation and on the main challenges of implementation. Governance issues include the capacity of health authorities to rule, reinforce and have control of the implementation of abortion policy for the provision and access to comprehensive abortion care. We will address the policy dynamics from the perspective of policymakers, health services managers and women's organisations.

In dimension 3 (service delivery arrangements) we will assess how health services are organised to ensure access to quality abortion care, and how barriers and facilitators are addressed by different strategies.[3 10–13]

In dimension 4 (health system and health outcomes indicators) we will assess what information on health system and health outcomes indicators related to abortion care are monitored and evaluated. We will address the process of generating, collecting, reporting, monitoring and evaluating health system and health outcomes

indicators, and how this evaluation accounts for policy implementation progress and quality of care oversight.

We will assess these dimensions through availability, accessibility, acceptability and quality of care as framed in the Availability, Accessibility, Acceptability and Quality of Care (AAAQ) model. We will use this framework to gather and systematise the data coming from the two data collection instruments: semi-structured interviews and document review. *Availability* refers to the need of having enough functioning public health and healthcare facilities, goods and services. *Accessibility* refers to the need of making all those facilities, goods and services physically accessible and affordable to everyone within the jurisdiction of the state party without discrimination. *Acceptability* refers to the need of the health system responses to be respectful of the law, medical ethics and to be culturally appropriate and sensitive to gender and age. *Quality of Care* refers to the need for health goods and services to be scientifically and medically approved and of good quality following WHO standards.[14] Specifically, *quality of abortion care* refers to the fact that the provision of abortion services should be effective, efficient, accessible, acceptable (patient centred), equitable and safe.[1]

As a result of the appraisal of the four dimensions, we will identify the **barriers and facilitators** to availability, accessibility, acceptability and quality of care regarding abortion related to abortion policy implementation.

Finally, we will synthesise findings from the case studies to draw lessons learnt on cross-cutting barriers and facilitators, as well as the similarities and differences on how comprehensive abortion policies are implemented in the participant countries. These lessons learnt can be used to inform future decision-making across the region and other sociocultural and geopolitical contexts (see figure 1).

For this study, comprehensive abortion care includes the elements of induced abortion under legal indications as well as all the elements of post-abortion care: information, counselling, treatment, contraceptive services, reproductive health services, community and service providers' partnerships.

## Settings

We will conduct case studies in five selected countries: Argentina, Colombia, Honduras, Mexico and Uruguay. The selection of settings addresses diversity in abortion regulations as well as diversity in countries' institutional political organisation across a continuum of abortion-related legal access. We aim to include countries going from an absolute legal restrictive environment with only post-abortion care provision to countries with less restrictive legal status of abortion, availability of abortion guidelines or registration and availability of mifepristone and misoprostol, two medicines used in the medical abortion process. The selection also considers common issues on abortion policy implementation and indicators among countries. Country profiles are summarised in table 1.

## Data collection and tools

We will use two data collection methods: desk review (for dimensions 1 and 4) and semi-structured interviews (for dimensions 1, 2, 3 and 4). The data collection process is summarised in table 2. We will use the quantitative data not only to inform the qualitative process, but to integrate it into the analysis to better elucidate different aspects of the dimensions, thereby allowing for a robust process of triangulation.

### Desk review

For dimension 1, we will review the texts of regulations to identify and categorise abortion-related regulations and we will structure a data extraction matrix into two domains:

▶ The abortion regulatory framework, which refers to the specific regulations and practices-related to abortion.
▶ The legal environment, which includes regulations that are not specifically about abortion but have a

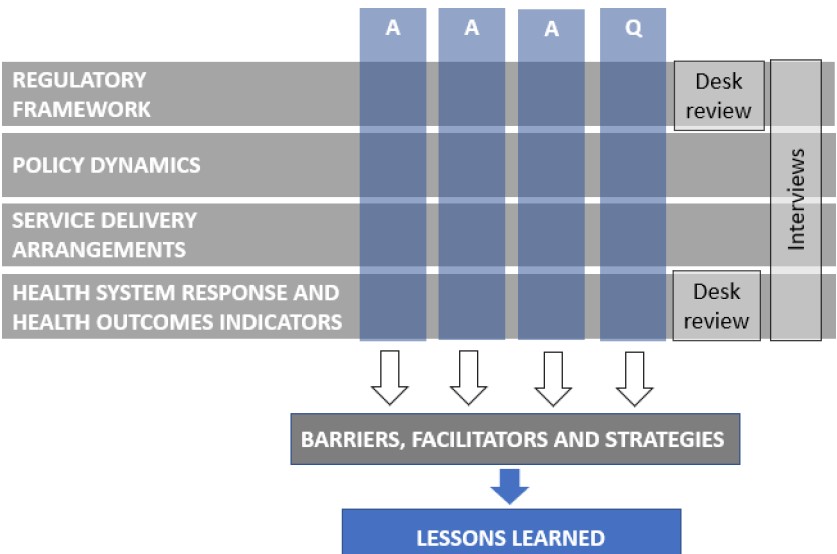

**Figure 1** Research framework: dimensions and methods. AAAQ, Availability, Accessibility, Acceptability and Quality of Care.

**Table 1** Country profiles

| | Argentina | Colombia | Honduras | Mexico* | Uruguay |
|---|---|---|---|---|---|
| Socio economic profile | ▲ 42% of the population lives under poverty.† <br> ▲ 42, 3 Gini coefficient.‡ <br> ▲ High level of inflation. The second country with highest (triple digit inflation). <br> ▲ A historically large middle class is shrinking. | ▲ 42% of the population lives under poverty.† <br> ▲ 54, 2 Gini coefficient.‡ <br> ▲ Under a peace agreement process, with still sites with high levels of violence and isolation. | ▲ 48% of the population lives under poverty.† <br> ▲ 48, 2 Gini coefficient.‡ <br> ▲ The size of the middle class was among the smallest in the region (18% compared with an average 41%). <br> ▲ High levels of violence: over 38 homicides per 100 000 inhabitants. | ▲ 44% of the population lives under poverty.† <br> ▲ 45, 4 Gini coefficient.‡ <br> ▲ High levels of social and political violence connected with the illegal drug market. | ▲ 11% of the population lives under poverty.† <br> ▲ 40, 3 Gini coefficient.‡ <br> ▲ Its middle class is the largest in the Americas, representing more than 60% of its population.§ |
| Division of the geographical power | Federalism with shared competences between national and local government. | Centralist with local powers. | Unitarian. | Strong federalism. | Centralist. |
| Abortion legal model | Mixed (indications and voluntary interruption of pregnancy) with partial decriminalisation. | Mixed with total decriminalisation up to 24 weeks. | Total ban. | Mixed at some states, indication model in other states. | Mixed with partial decriminalisation. |
| Main characteristics of abortion regulation | ▲ The law allows any health professional to provide abortion services. <br> ▲ No rape report is required for access legal abortion under the rape indication. <br> ▲ There is no abortion committee. <br> ▲ Conscientious objection is allowed but with limitations and only at an individual level. | ▲ The law allows any health professional to provide abortion services. <br> ▲ There is no abortion committee. <br> ▲ Ambulatory abortion is accepted. <br> ▲ No rape report is required for access legal abortion under the rape indication. <br> ▲ Conscientious objection is allowed but with limitations and only at an individual level. | ---------- | ▲ In some states rape report is required. <br> ▲ There is no abortion committee. <br> ▲ Conscientious objection is allowed but with limitations and only at an individual level. | ▲ There is an abortion committee. <br> ▲ Medical-based regulation, with several other requirements. <br> ▲ Conscientious objection is allowed in broad terms, both individual and institutional. |
| Protocol of safe and legal abortion | Yes. National. | Yes. National. | No. Just protocol for management of obstetrical complication. | Yes Locals with variations. | Yes. National. |
| Institutional adjudication | Congress. | Court. | ---------- | Local legislatures. | Congress. |
| Level where care is provided | First and second level. | First and second level. | Second level. | First and second level. | Second level. |
| Health service arrangements | Public health sector is the main abortion provider together with the community sector (also called acompañantes or informal health system). | Private clinics (Pro Familia and Orientame) are two of the main abortion providers. | Some public hospital are the ones proving abortion care. | Public specialised abortion clinics are the main providers in Mexico City. | Public health sector is the main provider, and private clinics are not promoted. |

*Mexico has different regulation in each state, so the references here are regarding the national level.
†World Bank. Pobreza en América Latina. Datos estadísticos 2022. Available at: https://es.statista.com/temas/8701/pobreza-en-america-latina/#editorsPicks.
‡World Bank. Desigualdad en la distribución de ingresos basado en el coeficiente Gini en América Latina y el Caribe en 2021, por país. 2021. Available at: https://es.statista.com/estadisticas/1267584/latinoamerica-coeficiente-gini-desigualdad-de-ingresos-por-pais/.
§Word Bank. Uruguay: panorama general. 2023. Available at: https://www.bancomundial.org/es/country/uruguay/overview#:~:text=Ent%C3%A9rminosrelativos%2Csuclase,laCOVID%2D19en2020.

**Table 2** Data collection

| | Dimension 1 | Dimension 2 | Dimension 3 | Dimension 4 | Total |
|---|---|---|---|---|---|
| Argentina | Desk review 20 interviews | 20 interviews | 20 interviews | Desk review 10 interviews | 2 desk reviews 70 interviews |
| Colombia | Desk review 20 interviews | 20 interviews | 20 interviews | Desk review 10 interviews | 2 desk reviews 70 interviews |
| Honduras | Desk review 20 interviews | 20 interviews | 20 interviews | Desk review 10 interviews | 2 desk reviews 70 interviews |
| Mexico | Desk review 20 interviews | 20 interviews | 20 interviews | Desk review 10 interviews | 2 desk reviews 70 interviews |
| Uruguay | Desk review 20 interviews | 20 interviews | 20 interviews | Desk review 10 interviews | 2 desk reviews 70 interviews |
| Total (five countries) | 5 desk reviews 100 interviews | 100 interviews | 100 interviews | 5 desk reviews 50 interviews | 10 desk reviews 350 interviews |

direct impact on access to abortion health services (eg, laws regarding patient confidentiality, norms regarding minors' consent, norms on mental or intellectual disabilities, laws related to sexual violence).

For dimension 4, we will conduct a systematic search for country-specific statistics, reports and documents. National official reports and databases containing processes, outputs or outcome indicators relevant to abortion care at national level will be reviewed and gaps in indicator reporting, monitoring and evaluation will be identified.

We will base the data extraction matrix for dimension 1, on the topics included in the WHO Global Abortion Policies Database[9 15] and in the report 'Barreras en el acceso a los abortos legales: una mirada a las regulaciones sanitarias que incluyen el uso del misoprostol'.[16] For dimension 4, we will use a data extraction matrix structured with the AAAQ domains to identify and categorise abortion-related indicators. From each reported indicator, we will collect data on indicator definition, method of measurement, data collection methodology and frequency, date of publication, source of information and indicator value. If data are stratified (eg, race, gender, age, urban/rural), this information will also be collected.

### Semi-structured interviews

We will conduct semi-structured interviews[17] to explore:

► How legal and regulatory instruments are used, how they pose as barriers and facilitators to abortion service provision and access and how regulations are translated into practice (dimension 1).
► The policy dynamics related to agenda setting, governance issues, financing schemes and actors' involvement (dimension 2).
► Which are the service arrangements adopted according to the different settings, human resources and the technologies available and which were the triggers for the model/different models currently used (dimension 3).
► Which approaches each country uses for monitoring abortion-related care and for the oversight

of maternal mortality, near miss and quality of care and how monitoring and evaluation of process—and outcomes—indicators related to abortion are operationalised (dimension 4).

An experienced interviewer with social science or public health background with legal or programmatic knowledge will conduct the interviews. The interviewers will obtain informed consent, conduct the interviews in person or through digital technology and record and transcribe the interviews.

### Participants

For the interviews, we will use a purposive sample of 20 key informants for dimensions 1–3 and 10 key informants for dimension 4 in each country. We seek to obtain a broad range of perspectives on the subject.[18] The selected subjects will be key informants in the abortion policy domain, encompassing different categories: health providers, sexual and reproductive health policy decision-makers, researchers, leaders of women's organisations, judicial agents, politicians, opinion leaders, etc.

The sample size is a target overall number, which we will adjust during the data collection process. We consider that 20 key informants for dimension 1–3 and 10 key informants for dimension 4 in each country is a feasible sample size that should allow for variability along the different categories (see table 2).

The identification of participants is expected to be different depending on how broad or narrow the categories or levels from where they are identified are. The criteria of variability of forums or levels aims at obtaining diversity and having those that yield the most plentiful data in the matter.

We will identify the key informants through a mapping of actors. A first round of mapping will be based on teams' expertise and networks. We expect to easily identify narrow-level participants such as decision-makers (who are few in each national abortion domain). For broad categories, such as health providers and activists (who are many and disperse in different settings), we will ultimately define the selection strategy with each country's

principal investigator to achieve a convenience sample that accounts for variability. There may be a second round of mapping to select knowledgeable stakeholders identified by the interviewees from the first round.

## Analysis

### Document review data analysis (each country study)

For dimension 1, we will chart and narratively synthesise data from the document review based on the explicit wording and content of the text, following the categories of the WHO Global Abortion Policies Database.[9 15] This WHO database contains information related to authorisation and service-delivery requirements, conscientious objection, penalties, national sexual and reproductive health (SRH) indicators and UN Treaty Monitoring Body concluding observations among other documents.

The written words of the documents, however, take their meaning from their purpose and context and often require interpretation. Therefore, for the interpretation of competing or inconsistent legal primary sources, legal secondary sources will be used. Additionally, we will carry out a preliminary classification as barrier or facilitator for each topic for the purpose to assess them in the interviews.

For dimension 4, we will synthesise data from the document review and further collect indicators for abortion care; we will classify and analyse this data according to the AAAQ domains. The data extraction matrix will also be based on indicators for abortion care as identified as part of the WHO 2022 indicators for abortion,[1] the Danish Institute for Human Rights reports on sexual and reproductive health and rights and the systematic review by Dennis *et al.*[19–22] We will identify gaps between the extraction matrix and the available indicators.

We will use data from the desk reviews to gain a comprehensive understanding of the subject matter and identify key themes or gaps in knowledge. We will also use country specific statistics and documents to inform the interviews.

### Interview data analysis (country studies)

We will use an inductive thematic coding approach with the data from the interviews for each dimension and organise a matrix round central aspects of the corresponding dimension. A content analysis will follow the coding to identify emerged themes and patterns among them.[23] We will identify barriers and facilitators of policy implementation in relation to the AAAQ framework in a data reduction matrix. The first step will be to extract the data from the interviews and place it within the topics of the data extraction matrix. Then, a second reading of the data in the matrix will take place, guided by specific questions for each dimension. Finally, the analysis of barriers, facilitators and strategies will be organised under the AAAQ matrix.

We will conduct a stakeholder analysis to map the actors who have had an interest and have affected or influenced the implementation of abortion policy (dimension 2), to better understand their role in the continuum supporters-blockers, following the questions suggested by Schmeer[24]:

► Who are the most important abortion stakeholders (from a power and leadership perspective)?
► What are the different stakeholders' positions on abortion policies?
► What do the different stakeholders see as possible advantages or disadvantages of abortion policies?
► Which stakeholders form alliances? Is there potential for new alliances to form?

### Identification and synthesis of barriers, facilitators and strategies of comprehensive abortion policy implementation (country studies)

We will then apply the findings from dimensions 1–4 to a data grid organised with the AAAQ framework to better illustrate the barriers, facilitators and strategies to deal with them. This aims to group the information from the four dimensions in one single document, which will shed light on strategies used towards comprehensive abortion policy implementation (figure 1).

We will analyse this information using substantive themes (or codes or clusters of codes). The rearrangements and recombination may be facilitated by depicting the data graphically or by arranging them in lists, for example.

The collection of quantitative data will provide a quantifiable framework to evaluate barriers and facilitators in abortion care access and quality. We will obtain a holistic view by integrating quantitative and qualitative data, addressing both the qualitative depth of policy experiences and the quantitative breadth of their impact on abortion care policy.

### Lessons learnt from the five country studies (regional synthesis)

At this final stage of the study, we will conduct a regional synthesis exercise to draw lessons on barriers, facilitators and the strategies provided by the five country studies. Once the first version of this synthesis exercise is ready, we will conduct a validation process involving members of all five national teams to reach consensus on the synthesis. For this validation, we will apply an adjusted nominal group technique.[25]

### Summary of study process

We expect to start the study in February 2024 and finish it in May 2025. The study process is summarised in table 3.

### Positionality statement

We are researchers working in the field of abortion policy implementation, sexual and reproductive rights and evidence-based practices, most living in Argentina. We will work on making sure that any biases or assumptions we have or we may bring are documented.

### Patient and public involvement

None.

**Table 3** Study process

| | Data collection | Data analysis | Final product |
|---|---|---|---|
| Argentina | D1–D4<br>2 desk reviews<br>70 interviews | D1–D4<br>Barriers and facilitators<br>Model AAAQ | One country report |
| Colombia | D1–D4<br>2 desk reviews<br>70 interviews | D1–D4<br>Barriers and facilitators<br>Model AAAQ | One country report |
| Honduras | D1–D4<br>2 desk reviews<br>70 interviews | D1–D4<br>Barriers and facilitators<br>Model AAAQ | One country report |
| Mexico | D1–D4<br>2 desk reviews<br>70 interviews | D1–D4<br>Barriers and facilitators<br>Model AAAQ | One country report |
| Uruguay | D1–D4<br>2 desk reviews<br>70 interviews | D1–D4<br>Barriers and facilitators<br>Model AAAQ | One country report |
| Total (five countries) | 10 desk reviews and 350 interviews | Barriers and facilitators per dimension in the five countries<br>Model AAAQ | Five country reports |
| Regional level | - | Synthesis exercise of the five country reports | One regional report on lessons learnt |

AAAQ, Availability, Accessibility, Acceptability and Quality of Care.

### Ethics and dissemination

The project has been approved by the WHO Research Ethics Review Committee (ID: A66023). Moreover, it has been approved by the local Research Ethics Committee in the five participating countries: Centro Rosarino de Estudios Perinatales (Argentina), Profamilia CEIP (Colombia), Universidad Autónoma de Honduras CEIB (Honduras), Instituto Nacional de Salud Pública (Mexico) and Centro Hospitalario Pereira Rossell (Uruguay).

The study will respect the ethical safeguards necessary to obtain informed and voluntary consent and preserve the anonymity of the key informants interviewed. Names of individuals and organisations will not be used in any final or publicly available product resulting from this research. Interviewers will be professionals with proven experience to guarantee ethical principles in conducting the interviews, date and time will be based on the participant's availability.

Data collected in this study will be managed, stored, analysed and results will be disseminated with careful attention to protecting human subjects' rights to privacy and confidentiality. Specifically, data will be treated in compliance with and approved by the WHO's Research Ethics Review Committee and Centro Rosarino de Estudios Perinatales Independent Ethics Committee.

Findings from the study will be disseminated through a multipurpose strategy to target diverse audiences so as to foster the use of the study findings to inform the public debate agenda and policy implementation at national level. The dissemination strategy will include:

► Peer-reviewed publication: Academic article(s) will be submitted to international journals to disseminate main findings of the comparative analysis of barriers and facilitators of quality abortion policy implementation in the selected countries.
► National dissemination workshop: This workshop will gather diverse political, academic and policy actors to discuss main findings and discuss barriers and facilitators of quality abortion policy implementation.
► Regional dissemination workshop: This workshop will gather diverse political, academic and policy actors to discuss main findings of the comparative analysis of barriers and facilitators of quality abortion policy implementation in Latin America.

### DISCUSSION

Latin American countries have several legal and other restrictive conditions regarding abortion. However, some countries of this region also have legal norms that could allow a better access to quality abortion care but are not implemented or are ill-implemented. Interestingly, in the last years, several strategies have been adopted, which have led to improved access to abortion care, such as broad interpretation of existing regulatory frameworks and of policy governance reinforcement, as well as social pressures exerted on decision-makers and politicians by women's organisations.

The WHO Global Abortion Policies Database provides a comprehensive compilation of country-specific documents regarding policies and laws.[9 15] Existing literature has explored abortion policy implementation in various contexts, including a comparative case study that investigated health sector strategies that were useful in expanding or establishing abortion services in six countries around the world.[26] Context and actors involved in abortion reform in some Latin-American countries have

also been described,[27 28] with a recent cross-sectional study characterising abortion-related complications, management of these complications and women's experiences with abortion in six countries.[29] A strength of this study is the assessment of different dimensions and actors of abortion policy implementation, which should provide a comprehensive and more precise understanding of the factors that influence the implementation of these types of health policies. Nevertheless, from all complex social, legal, cultural and ethical factors shaping abortion care and policies, this research will focus on only four to identify relevant barriers and facilitators to abortion policies, using the AAAQ framework.

Another strength of this project is that, as a multi-country study, it encompasses five countries from restrictive to more liberalised environments. This will show different contexts while looking at common patterns regarding barriers, facilitators and strategies. This data will be collected through standardised instruments and validated by the teams involved in the project. However, there are challenges in providing a systematic comparison of heterogeneous countries. Moreover, the samples in this study may not reflect the heterogeneity of institutions and the proportionality of the population. To capture this complex scenario in each of the five country-case studies, local interview guides will be adjusted and expanded based on contextual factors.

This study will focus on factors influencing implementation and will not analyse how these factors might correspond to other aspects, such as the evaluation of abortion policies, the estimation of satisfaction with the care provided or the magnitude and severity of abortion-related complications in relation to policy implementation. Future research may consider including these and other aspects.

Finally, principal and country investigators will commit to include junior researchers in the development of the study. By conducting this research project, the coordinating team and the local teams will strengthen their skills, including those of junior researchers, to provide more responsive and sustainable policies that address women's needs and rights.

**Author affiliations**
[1]Centro Rosarino de Estudios Perinatales (CREP), Rosario, Argentina
[2]Blanquerna Ramon Llull University Faculty of Health Sciences, Barcelona, Spain
[3]Centro de Estudios de Estado y Sociedad (CEDES), Buenos Aires, Argentina
[4]Consejo Nacional de Investigaciones Científicas y Técnicas, (CONICET), Buenos Aires, Argentina
[5]Latin American Center of Perinatology Women and Reproductive Health (CLAP/WR), Pan American Health Organization, Montevideo, Uruguay
[6]Department of Global Public Health, Karolinska Institutet, Stockholm, Sweden
[7]Department of Sexual and Reproductive Health and Research and UNDP-UNFPA-UNICEF-WHO-World Bank Special Programme of Research, Development and Research Training in Human Reproduction (HRP), World Health Organization, Geneva, Switzerland

**Acknowledgements** We thank the country principal investigators for their comments to the protocol and contributions regarding country specific information: Edgardo Abalos (Argentina), Rocío Murad (Colombia), Jakeline Álger (Honduras), Karla Berdichevsky (Mexico), Karla Flores (Mexico) y Cecilia Stapff (Uruguay). We also thank Agostina Allori for reviewing dimension 1 and Daniel Giordano for his inputs regarding data management and data security aspects.

**Contributors** AL, CG, ARM, SR, GC, MR contributed to the conception and design of the study protocol, which was discussed with the rest of the authors. CG, ARM, SR, GC, MR, BC, RGPdL and AL agreed on the methodological aspects and participated in developing the protocol, reviewing the draft, and making substantial revisions. CG wrote the original outline. ARM together with CG were responsible for the writing, reviewing and editing of the subsequent versions of the protocol. ARM oversaw developing dimension 1 section. SR was in charge of developing dimension 2. MR was in charge of developing dimension 3. GC and CG were in charge of dimension 4. BC developed and wrote all ethics considerations and prepared the original figures with SR and CG. BC and MVO revised it for intellectual content and provided feedback. CG and ARM led the manuscript writing. CG and ARM, with the help of MVO, coordinated the steps for article development and wrote the draft of the article together. AL, SR, GC, MR and RGPdL contributed to manuscript writing and revision. All authors contributed to read and approved the submitted version.

**Funding** This project is funded by the UNDP-UNFPA-UNICEF-WHO-World Bank Special Programme of Research, Development and Research Training in Human Reproduction (HRP), Department of Sexual and Reproductive Health and Research, World Health Organization, 20 Avenue Appia, 1211 Geneva, Switzerland. The content of this article is solely the responsibility of the authors and does not reflect the views of the referred programme.

**Competing interests** None declared.

**Patient and public involvement** Patients and/or the public were not involved in the design, or conduct, or reporting, or dissemination plans of this research.

**Patient consent for publication** Not applicable.

**Provenance and peer review** Not commissioned; externally peer reviewed.

**ORCID iDs**
Celina Gialdini http://orcid.org/0009-0004-1150-2142
Rodolfo Gomez Ponce de León http://orcid.org/0000-0001-9914-7130

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
