## [Reviewer comments · BMJ Open]

ARTICLE DETAILS

TITLE (PROVISIONAL)	Multi-country research on comprehensive abortion policy implementation in Latin America: a mixed-methods study protocol
AUTHORS	Gialdini, Celina; Ramón Michel, Agustina; Romero, Mariana; Ramos, Silvina; Carroli, Guillermo; Carroli, Berenise; Gomez Ponce de León, Rodolfo; Vila Ortiz, Mercedes; Lavelanet, Antonella

VERSION 1 – REVIEW

REVIEWER	Keogh, Louise The University of Melbourne
REVIEW RETURNED	20-Jun-2023

GENERAL COMMENTS	This protocol describes a comprehensive study design to understand the regulation, policy and indicators of abortion services in five countries in Latin America with differing legal and healthcare settings. The study design appears robust, and likely to produce useful results. However, I suggest the authors attend to the following minor issues before publication. The five countries: Authors should justify the choice of the five countries earlier in the paper. Where more detail is provided under "setting" can the conditions in each country be provided in table for comparison and to show the spread of contexts that will be studied. Qualitative interviews: The text suggests 80 participants per country, which is 400 interviews (table suggests 70 - please remove discrepancy). Either way, this is a very large number of interviews and will be onerous to analyse - further justification as to why so many interview are required and the feasibility of analysis should be provided. For example - it is likely that one participant will be able to comment on more than one aspect of the AAAQ, therefore you may not need 20 participants per aspect in each country. Aims / Research question: Aims are provided - can the authors also identify a research question? Dimensions: Regulation is well defined initially, however in the data collection section, policy is introduced to dimension 1. Consider where policy best sits, and clearly define what constitutes 'policy' versus regulation early in the paper. AAAQ model should be defined early in the paper
---

	sentence line 13-18 p12 doesn't make sense - please amend
--	---

REVIEWER	LaRoche , Kathryn Purdue University
REVIEW RETURNED	29-Aug-2023

GENERAL COMMENTS	Thank you for giving me the opportunity to review this study protocol that aims to document and understand the implementation of abortion policy in five Latin American countries. This is an exciting and well thought out study, and the article is well written. However, below I raise some small points that could strengthen the study protocol:  1. This study uses two methods that rely on qualitative data; therefore, this is a multi-methods study, not a mixed methods study. Mixed methods implies the integration of qualitative and quantitative data which does not come through in the manuscript. 2. Given the qualitative nature of this study protocol, it would be consistent with best practice in qualitative approaches to write the article in active voice. Qualitative research specifically acknowledges the role of the researcher in the research process, and this change would help improve readers' understanding of the topic. For readers less familiar with abortion, it might be helpful to explain what mifepristone and misoprostol are, and to explain what is considered PAC earlier than it appears on P7. The protocol says that they will "systematically assess comprehensive abortion policies" in five countries, and also says it aims to improve the "implementation of abortion policies." Being more specific here would be helpful. Are the authors considering both pro-abortion and anti-abortion policies in their review? Will stakeholders who are anti-abortion be included in the study? And finally, I urge the authors to be specific that they are looking at strategies to improve access to evidence-based abortion care. I understand that it is beyond the scope of this project to include the voices of women and abortion seekers, but I think it's flawed to say that women's organizations could capture this sentiment. The study protocol is strong and sufficient without including the voices of abortion patients, but the authors should be careful not to conflate perspectives from organizations with the perspectives and experiences of actual abortion seekers. I would also like to see more information about the proposed analytic process as it is currently quite vague. Given the focus on laws and policies in this study, does the study team have someone with legal expertise on board to help with the interpretation? Copy-editing notes: P4, L54: "Despite the efforts and claims, abortion laws have been difficult to reform." This sentence is difficult to interpret and "efforts and claims" is an unclear word choice. P5, "twelve-fourteen weeks" - this sentence is discussing abortion laws, and I lack clarity about when abortion is permitted
---

	P4 and 5: It would be helpful to add additional information about time periods to help contextualize your study protocol. When the authors say “Currently abortion is legal without reason ...” please add dates/time period. As well, add a timeframe when talking about the Columbian Constitutional Court decision. P5, L15: “... resorting to other instruments”. Instruments seems like the wrong word – maybe "strategies" would be better? Once again, thank you for giving me the opportunity to review this paper. Note: I believe in a transparent review process and I have provided the same comments to the authors and the editor(s).
--	---

VERSION 1 – AUTHOR RESPONSE

Reviewer 1 comments	Authors' response
This protocol describes a comprehensive study design to understand the regulation, policy and indicators of abortion services in five countries in Latin America with differing legal and healthcare settings. The study design appears robust, and likely to produce useful results. However, I suggest the authors attend to the following minor issues before publication.	Thank you for your revision and for taking the time to send these comments and suggestions.
The five countries: Authors should justify the choice of the five countries earlier in the paper. Where more detail is provided under "setting" can the conditions in each country be provided in table for comparison and to show the spread of contexts that will be studied.	Thank you. A table has been added
Qualitative interviews: The text suggests 80 participants per country, which is 400 interviews (table suggests 70 - please remove discrepancy). Either way, this is a very large number of interviews and will be onerous to analyse - further justification as to why so many interview are required and the feasibility of analysis should be provided. For example - it is likely that one participant will be able to comment on more than one aspect of the AAAQ, therefore you may not need 20 participants per aspect in each country.	Thank you. It was edited in lines 325 and 334 and discrepancy was removed. Dimension 4 has 10 IDIs. Regarding the number of interviews, it is worth mentioning that each country team will oversee its own data collection and analysis. Within each country team a dimension investigator will conduct and analyze 20 interviews (or 10 for dimension 4). In any case, as suggested in line 267, 20 key informants per dimension is an approximate number that will be adjusted during the data collection process. (In case responses from one key informant addresses issues

	of more than one dimension the number of final interviews might be adjusted).
Aims / Research question: Aims are provided - can the authors also identify a research question?	The protocols approved by all ERCs contained the following research question. How do context, content, actors and processes affect the implementation of comprehensive abortion policies? We have added it in the introduction section (line 129-130)
Dimensions: Regulation is well defined initially, however in the data collection section, policy is introduced to dimension 1. Consider where policy best sits, and clearly define what constitutes 'policy' versus regulation early in the paper.	Thank you for pointing this inconsistency. Policy is dimension 2, it was removed from dimension 1. We have added definitions of 'policy' and 'regulation' after line 166.
AAAQ model should be defined early in the paper	The core of our methods are the 4 dimensions, which in turn will be assessed through the AAAQ model. That is why we first present the dimensions and only therefore the AAAQ framework. However, we have added some references early in the paper (line 111).
sentence line 13-18 p12 doesn't make sense - please amend	We could not identify the sentence line as the number line does not seem to match the page. We would be happy to address as needed with more information.
Reviewer 2	Authors' response
Thank you for giving me the opportunity to review this study protocol that aims to document and understand the implementation of abortion policy in five Latin American countries. This is an exciting and well thought out study, and the article is well written. However, below I raise some small points that could strengthen the study protocol: 1. This study uses two methods that rely on qualitative data; therefore, this is a multi-methods study, not a mixed methods study. Mixed methods implies the integration of qualitative and quantitative	Thank you very much for your comments, which helped us improve the manuscript. While we recognize that our methodology may appear as multi-method, we will use the quantitative data to both inform the qualitative process and the analysis to better elucidate different aspects of the dimensions of the study. The quantitative aspects are not

data which does not come through in the manuscript.	merely descriptive. This has been clarified in the text in the “Methods” section (lines 263-265, and 387-390).
2. Given the qualitative nature of this study protocol, it would be consistent with best practice in qualitative approaches to write the article in active voice. Qualitative research specifically acknowledges the role of the researcher in the research process, and this change would help improve readers' understanding of the topic.	Thank you very much. We have made the corresponded changes.
For readers less familiar with abortion, it might be helpful to explain what mifepristone and misoprostol are, and to explain what is considered PAC earlier than it appears on P7.	Thank you. We have included a short reference of these two medications as standard medication for having safe abortions. Please see line 221-222. We have also included the complete name of PAC.
The protocol says that they will "systematically assess comprehensive abortion policies" in five countries, and also says it aims to improve the "implementation of abortion policies." Being more specific here would be helpful. Are the authors considering both pro-abortion and anti-abortion policies in their review? Will stakeholders who are anti-abortion be included in the study? And finally, I urge the authors to be specific that they are looking at strategies to improve access to evidence-based abortion care.	Thank you for the suggestion. In the paper we included the notion of comprehensive as a broader term that encompass safe abortion policies and health care and PAC. Regarding the pro-abortion/anti-abortion: the selection criteria for interviewees/stakeholder will take into consideration their role, and not their personal position in favor of or against abortion. In line with your recommendation, we have added this specification: strategies to improve access to evidence-based abortion care. Please see lines 69, 137.
I understand that it is beyond the scope of this project to include the voices of women and abortion seekers, but I think it's flawed to say that women's organizations could capture this sentiment. The study protocol is strong and sufficient without including the voices of abortion patients, but the authors should be careful not to conflate perspectives from organizations with the perspectives and experiences of actual abortion seekers.	We thank you for this comment. We agree on both points: the protocol is strong enough without patients' voices and organizations should not be taken as a proxy. We have eliminated this from the discussion section.
I would also like to see more information about the proposed analytic process as it is currently quite vague.	Thank you. We recognize that analysis is not described in depth. We decided for this stage to describe it in general terms that the quantitative data will not only inform the qualitative

	process, but will be integrated into the analysis to better elucidate different aspects of the dimensions, thereby allowing for a robust process of triangulation.
Given the focus on laws and policies in this study, does the study team have someone with legal expertise on board to help with the interpretation?	Both the regional team coordinating the study and the country teams will have a person with advanced legal training and knowledge in charge of dimension 1. This person is expected to have experience in the analysis of legal documents regarding health topics, preferably in the field of sexual and reproductive health. This person will also provide legal support if any members of the team should require so. Moreover, one of the Co-PI of the study is a legal expert in the field of sexual and reproductive health.
Copy-editing notes: P4, L54: “Despite the efforts and claims, abortion laws have been difficult to reform.” This sentence is difficult to interpret and "efforts and claims" is an unclear word choice. P5, "twelve-fourteen weeks" - this sentence is discussing abortion laws, and I lack clarity about when abortion is permitted P4 and 5: It would be helpful to add additional information about time periods to help contextualize your study protocol. When the authors say “Currently abortion is legal without reason ...” please add dates/time period. As well, add a timeframe when talking about the Columbian Constitutional Court decision. P5, L15: “... resorting to other instruments”. Instruments seems like the wrong word – maybe "strategies" would be better?	Thank you very much for noting this. We have made the changes to improve these sentences.

VERSION 2 – REVIEW

REVIEWER	Keogh, Louise The University of Melbourne
REVIEW RETURNED	08-Nov-2023
GENERAL COMMENTS	Thanks to the authors for changes made in response to reviewers.

	The addition of Table 1 is helpful, however, suggest the formatting/wording could be improved for publication In places the text is a little hard to follow, especially in relation to the steps of data analysis, hopefully copy editing prior to publication can ensure the steps are very clear, and that which datasets are being analysed at which step is also clear.
--	---

REVIEWER	LaRoche, Kathryn Purdue University
REVIEW RETURNED	14-Nov-2023

GENERAL COMMENTS	Thank you for giving me the opportunity to review a revised version of this manuscript. The authors have done a good job of addressing previous feedback from the reviewers and I thank them for the attention to detail they've had in making these changes. I have one remaining concern about the manuscript in its current form. The entire methods section needs to be recast for active voice. In its present form, the authors use active voice sometimes, but a lot of the section is written in passive voice which makes it hard to interpret and less precise. For example, the authors say "Semi-structured interviews will explore ..." which doesn't make sense because interviews can't do anything. They are a tool used to explore topics. As well, instead of saying "Dimension 1 will ...", it would be better to say "In Dimension 1, we will ..." Using active voice acknowledges the role of the researchers in the process of collecting and analyzing data which is critical for qualitative methods, and anything rooted in a post-positivist/interpretivist paradigm more broadly. - P4, L149: "mixed-method" should be "mixed-methods". However, I am not entirely convinced that this is a mixed-methods study – but rather a multi-method one. All of the dimensions described in the manuscript seem to be qualitative in nature. - In this vein, this is a qualitative project so when I was reading, I was looking for a positionality statement from the team. I understand that this is a protocol so the opportunity for reflexivity is limited at this point. However, it would strengthen the manuscript to include a statement about the positionality of the core PIs, and elucidate a strategy for hiring and training team members. Once again, thank you for giving me the opportunity to review a revised version of this manuscript.
---

VERSION 2 – AUTHOR RESPONSE

Comments from reviewer 1	Author's response
The addition of Table 1 is helpful, however, suggest the formatting/wording could be improved for publication	Thanks. Noted.
In places the text is a little hard to follow, especially in relation to the steps of data analysis, hopefully copy editing prior to publication can ensure the steps are very clear, and that which datasets are being analysed at which step is also clear	Thank you. We've gone through the text for grammar fixing and clarity.

Comments from reviewer 2	Author's response
I have one remaining concern about the manuscript in its current form. The entire methods section needs to be recast for active voice. In its present form, the authors use active voice sometimes, but a lot of the section is written in passive voice which makes it hard to interpret and less precise. For example, the authors say "Semi-structured interviews will explore ..." which doesn't make sense because interviews can't do anything. They are a tool used to explore topics. As well, instead of saying "Dimension 1 will ...", it would be better to say "In Dimension 1, we will ..." Using active voice acknowledges the role of the researchers in the process of collecting and analyzing data which is critical for qualitative methods, and anything rooted in a post-positivist/interpretivist paradigm more broadly	Thank you. Passive voice was changed to active voice.
P4, L149: "mixed-method" should be "mixed-methods". However, I am not entirely convinced that this is a mixed-methods study – but rather a multi-method one. All of the dimensions described in the manuscript seem to be qualitative in nature.	Thank you. Our approach to gathering indicators doesn't solely rely on qualitative data; we also prioritize the collection and analysis of quantitative data. Through meticulous gathering of statistics and precise figures, our aim is to offer a comprehensive, data-driven understanding of abortion policy implementation. This collection of quantitative data plays a pivotal role not only in informing interviews (Lines 306-308), but also in providing a quantifiable framework to evaluate barriers and facilitators in abortion care access and quality. The synergy between quantitative and qualitative data allows us to obtain a holistic view, addressing both the qualitative depth of policy experiences and the quantitative breadth of their impact on abortion care policy. We have strengthened the text explaining this use of indicators in our study. (Lines 341-344)

In this vein, this is a qualitative project so when I was reading, I was looking for a positionality statement from the team. I understand that this is a protocol so the opportunity for reflexivity is limited at this point. However, it would strengthen the manuscript to include a statement about the positionality of the core PIs, and elucidate a strategy for hiring and training team members	Thank you. We have included a positionality statement. (Lines 357-360) and we will explore this when recruiting and training country team members.
--	---